# TCAD Simulation of Resistive Switching Devices: Impact of ReRAM Configuration on Neuromorphic Computing

**DOI:** 10.3390/nano14231864

**Published:** 2024-11-21

**Authors:** Seonggyeom Kim, Jonghwan Lee

**Affiliations:** Department of System Semiconductor Engineering, Sangmyung University, Cheonan 31066, Republic of Korea

**Keywords:** neuromorphic computing, memristor, TCAD (technology computer-aided design), ReRAM (resistive random-access memory), KMC (kinetic Monte Carlo)

## Abstract

This paper presents a method for modeling ReRAM in TCAD and validating its accuracy for neuromorphic systems. The data obtained from TCAD are used to analyze the accuracy of the neuromorphic system. The switching behaviors of ReRAM are implemented using the kinetic Monte Carlo (KMC) approach. Realistic ReRAM characteristics are obtained through the use of the trap-assisted tunneling (TAT) model and thermal equations. HfO_2_-Al_2_O_3_-based ReRAM offers improved switching behaviors compared to HfO_2_-based ReRAM. The variation in conductance depends on the structure of the ReRAM. The conductance extracted from TCAD is validated in the neuromorphic system using the MNIST (Modified National Institute of Standards and Technology) dataset.

## 1. Introduction

Numerous new technologies such as big data, cloud computing, machine learning, and artificial intelligence are being developed and used. With technological advancements, the volume and complexity of data increases. A wide variety of intelligent applications are now mostly based on neural networks [1,2]. Neural network-based systems require large amounts of data and heavy computation for learning and deriving inferences. Therefore, it is necessary to develop memory devices based on new operating principles, innovative structures, and new materials [3]. Recently, there has been a focus on research into energy- and space-efficient devices. These include ultra-flexible origami tessellations with shape memory properties that can be integrated with electronic devices, and organic electrolyte transistors that can perform sensing, memory, and processing functions [4,5]. Traditional von Neumann computing-based systems are powerful for logical computations, but not efficient for neural network computations [6,7]. This is because the implementation of efficient neural networks requires both parallel synapse storage and computation [8]. The neuromorphic computing system (NCS) has been proposed as a solution to enhance the efficiency of neural network implementation. The NCS is a system that mimics the human brain. It is characterized by low power consumption and high-efficiency processing. Moreover, it enables the coexistence of data processing and storage. Research on NCSs is exploring two-terminal systems using memristors and crossbar structures, and three-terminal systems using ion-gated vertical transistors (IGVTs), carbon-based nanomaterials such as carbon nanotubes (CNTs) and graphene, and polymers [9,10,11]. NCSs use a memristor device as the synaptic unit and a crossbar structure for parallel computation [12]. Memristors are known to be suitable for neuromorphic systems due to their high-speed operation and low power consumption [13].

The memristor is a compound word for a memory device and a resistor, meaning a device that can serve as a memory device and a resistor [14]. Furthermore, it is categorized by material, including resistive random-access memory (ReRAM), phase-change random-access memory (PCRAM), and ferroelectric random-access memory (FeRAM). ReRAM is a memory that utilizes changes in resistance, storing “1” for low resistance and “0” for high resistance. The strengths of ReRAM include low operating voltage, low power consumption, high speed, and high density [15]. However, there are unresolved issues regarding materials, stability, and storage mechanisms [16]. PCRAM is a nonvolatile memory technology that exploits phase changes in materials for memory. Depending on the crystalline state of a material, it can switch between a crystalline state (“1” state) and an amorphous state (“0” state). The crystalline state is characterized by high optical reflectivity and low resistivity, while the amorphous state exhibits low optical reflectivity and high resistivity. PCRAM has the advantages of rapid read access, high density, and nonvolatility [17]. Meanwhile, it has the disadvantages of slower switching speed and higher power consumption than other memristor devices [18]. FeRAM displays nonvolatile resistive switching when the ferroelectric polarization direction of films reverses [19]. FeRAM offers nonvolatility, low power consumption, and fast operation, but faces challenges such as manufacturing difficulty, limited data retention time, and small memory capacity [20].

ReRAM is considered a promising device for modeling the features of biological synapses [21,22]. Although ReRAM has advantages in cell size and multi-bit capability, it has a higher energy consumption per bit than organic transistor, but smaller compared to conventional devices [23]. Furthermore, the ReRAM manufacturing process is similar to complementary metal-oxide semiconductors (CMOSs), so it does not incur any additional costs [24]. In particular, filament-based ReRAM provides low-power operation and excellent scalability [25]. In this study, the Synopsys’s Sentaurus technology computer-aided design (TCAD) is used to develop HfO_2_-based multilayer-structured ReRAM and identify its characteristics. The conductance is an important factor that affects accuracy in neuromorphic computing [15,24], and different types of conductance are verified depending on structures and input pulses. The accuracy is verified with NeuroSim using the MINST (Modified National Institute of Standards and Technology) data [26].

## 2. ReRAM Features

### 2.1. ReRAM Device Physics

Typically, the structure of ReRAM has metal–insulator–metal (MIM), where an insulator is sandwiched between metal electrodes. Oxide-based ReRAM devices operate based on redox reactions [24]. In ReRAM devices, redox reactions result in the formation of filaments (bridges) in the insulating layer between two metal electrodes. Memory storage has “0” and “1” states. A value of 0 means no data are stored, and a value of 1 means data are stored. The formation/rupture of the filaments induces the connection and disconnection of conductive paths between the two metal layers. This leads to the transition between low-resistance and high-resistance states [27,28]. A switch from HRS to LRS is called the SET process, and that from LRS to HRS is called the RESET process.

In most cases, a formation voltage greater than the SET voltage is required to induce a resistive switching operation in the initial device, and this operation is known as the forming process [29,30]. Figure 1 shows the operation of the RESET and SET processes. The switch between HRS and LRS is explained by the formation and rupture of conductive filaments within the insulator. These filaments comprise oxygen vacancies (V_o_) or metal deposits. It is proposed that the movement of oxygen ions/vacancies stimulated by Joule heating and electric fields is critical for switching [31].

### 2.2. ReRAM Materials

Materials used in ReRAM devices include Pt, Au, and TiN for electrochemically inert electrodes, Ti, Ni, Ta, Al, and Cu for active metal electrodes, and CuO_x_, TaO_x_, AlO_x,_ SiO_x_, ZrO_x_, TiO_x_, and HfO_x_ for insulating layers [32,33]. Oxide-based ReRAM operates based on the movement of anions, wherein oxygen vacancies form a conductance pathway within the insulating layer. Typically, an electrode that absorbs oxygen is required to facilitate anion movements [34]. HfO_2_ has been widely used as ReRAM materials due to its excellent CMOS compatibility. In HfO_2_-based ReRAM devices, TiN is commonly used as the electrode. TiN is utilized as an oxygen scavenger to remove oxygen ions from the HfO_2_ layer and functions as an oxygen reservoir. Most conductive filaments in HfO_2_ are composed of oxygen vacancies and are commonly used due to their high dielectric constant and excellent thermal and chemical stabilities. The switching characteristics of ReRAMs are more stable when combined with active metal electrodes, compared to HfO_2_ monostructures, due to the oxygen adsorption effect [35,36]. It is also reported that using a multilayer structure improves reliable switching and durability compared with employing a single insulating layer [37,38,39].

## 3. TCAD Modeling

The simulation utilizes the Sentaurus TCAD of Synopsys. TCAD is a tool used for developing and optimizing semiconductor processing technologies and devices via computer simulations. TCAD is widely used in the semiconductor industry. Despite its complex process technology, the use of TCAD reduces development costs and accelerates research. The widely recognized physical schematic of ReRAM switching is shown in Figure 2 [40].

The switching properties are related to the geometry of filaments owing to the direct result of the generation and recombination of oxygen vacancies within the insulating layer [40,41]. In the SET process, oxygen ions are removed from the lattice and migrate to the active electrode, leaving behind a conductive filament formed from oxygen vacancies, which reduces resistance. During the RESET process, a negative (−) voltage is applied, causing oxygen ions stored at the electrode–dielectric junction to migrate to the insulating layer, where they recombine with oxygen vacancies and break the filament, increasing resistance [31,41]. To describe the switching of oxide-based ReRAM, the behaviors of electrons and ions must be considered, and their random nature must be reflected upon. To reflect their random nature, the kinetic Monte Carlo (KMC) model is used. Figure 3 shows the SET–RESET mechanism in TCAD.

The oxygen ions are defined as *Particle1* in TCAD, representing non-conductive defects. This is applicable to diffusion and generation/recombination events. The oxygen ions and vacancies generate Frenkel defects during diffusion. The vacancies can diffuse within insulator and can be converted back into oxygen ions through recombination events. The vacancies are defined as *Particles2* in TCAD. Additionally, growth/recession events are added to vacancy. When sufficient vacancies diffuse, the conductive defects, known as filaments, are converted through growth events. When a recession event occurs, the filament is transformed back into vacancies. To simulate ReRAM in TCAD, SDevice must satisfy several conditions. First, we define traps, KMC defects, and particle/filament in the global *Physics* syntax. Next, KMC defects are set in the material *Physics* syntax based on the material, and the events are set as shown in Table 1.

The event rate is expressed as follows
(1)$$r=v\;exp\left[-\frac{E_{A}-pF}{k_{B}T}\right]$$
where v is the maximum rate of events, $E_{A}$ is the activation energy, p represents a molecular dipole, T is the temperature, $k_{B}$ is Boltzmann constant, and F denotes an electric field. For generation/recombination events occurring in the bulk, Frenkel pairs should be defined in the same domain, while for an interface, they should be defined in two different domains [42]. The KMC model provides trap-assisted-tunneling (TAT) model and a steady-state heat equation. The two models can be used to obtain realistic ReRAM characteristics. The TAT model utilizes Poole–Frenkel emission, which describes the emission of electrons from the conduction band of an insulator. The KMC Poole–Frenkel emission rate $R_{PF}$ is given by [42]
(2)$$R_{PF}=v\cdot \exp\left(-\frac{E_{D}}{k_{B}T}\right)\left\{{\left(\frac{k_{B}T}{\beta \sqrt{F}}\right)}^{2}\left[1+\left(\frac{\beta \sqrt{F}}{k_{B}T}-1\right)\exp\left(\frac{\beta \sqrt{F}}{k_{B}T}\right)\right]+\frac{1}{2}\right\}$$
where F denotes the insulator electric field, v is the lattice vibration frequency, $E_{D}$ represents the trap depth, $\varepsilon_{opt}$ is the insulator optical permittivity and $\beta =\sqrt{\frac{e^{3}}{\pi \varepsilon_{0}\varepsilon_{opt}}}$, with the electric charge e and the permittivity of free space $\varepsilon_{o}$. Table 2 represents the device characterization parameters. These parameters affect the ReRAM event and TAT. The heat equation is represented by [42]
(3)$$\vec{\nabla }\cdot \left(\kappa \vec{\nabla }T+\psi \sigma \vec{\nabla }\psi \right)=0$$
where ψ is a potential, κ and σ are electrical and thermal conductivities, which are spatially dependent considering the presence of defects.

## 4. NeuroSim Simulation

Ideally, the weight increase in long-term potentiation (LTP) and decrease in long-term depression (LTD) should be linearly proportional to the number of input pulses. However, real-world devices are not ideal and typically exhibit a sharp change in conductance in the early stages of LTP and LTD, followed by gradual saturation. NeuroSim uses a model that can capture nonlinear weight-update operations, and the change in conductance with the number of pulses is described by [26]
(4)$$G_{LTP}=B\left(1-e^{-\frac{P}{A}}\right)+G_{min}$$
(5)$$G_{LTD}=-B\left(1-e^{\frac{\left(P-P_{max}\right)}{A}}\right)+G_{max}$$
(6)$$B=(G_{max}-G_{min})/\left(1-e^{-\frac{P_{max}}{A}}\right)$$
where P is number of pulses. $G_{LTP}$ and $G_{LTD}$ represent the conductance of LTP and LTD, respectively. $G_{max}$ and $G_{min}$ indicate the maximum and minimum conductance, respectively. $P_{max}$ is defined as the maximum number of pulses required for the device to transition from minimum to maximum conduction. These values are directly extracted from experimental data. A is a variable that controls the nonlinear operation of weight updates. A of LTP and LTD is obtained by fitting in MATLAB R2022a. B is fitted within the ranges the $G_{max}$, $G_{min}$, $P_{max}$ and A. Upon completing fitting, the normalized value of A can be verified to identify the nonlinearity label, as shown in Figure 4, which is an example of fitting ReRAM weight updates. The actual conductance varies from cycle to cycle. This variation will be reflected in the weight update, resulting in a rough adjustment. NeuroSim provides nonlinearity label values that correspond to the normalized A values in the Nonlinearity-NormA.htm file. The nonlinearity label A provides a sufficiently precise value at 0–9 in 0.01 increments. If the A is negative, a negative value can be used for the nonlinearity label as well. The A obtained here are used in NeuroSim simulation.

The conductance of each structure is extracted, and the accuracy of the extracted values is verified with NeuroSim. It is developed to mimic an online/offline learning classification scenario using the MNIST handwriting dataset in a multilayer perceptron (MLP) neural network [26]. As shown in Figure 5, this neural network comprises an input layer, a hidden layer, and an output layer.

MLP is a fully connected neural network in which every neuron in each layer is connected to every neuron in the next layer. Connections of neurons are made through synapses, and the strength of a synapse represents its weight. $W_{IH}$ and $W_{HO}$ are weight matrices between the input and hidden layers and hidden and output layers, respectively. The input image data are an MNIST handwritten digit and comprise 20×20 pixels. The topology of the network is 400 (input layer)–100 (hidden layer)–10 (output layer). A total of 400 neurons in the input layer corresponds to a 20 × 20 MNIST image, and 10 neurons in the output layer represent 10 classes of numbers. Users can change the network topology as needed.

## 5. Result and Discussion

The ReRAM device proposed in this study has a Ti layer between the TiN top and bottom electrodes and an active layer comprising one or two insulators. As shown in Figure 6a, the basic structure comprises a 1 nm thick HfO_x_ and a TiO_x_ layer inserted between a 10 nm-thick HfO₂ layer and a 5 nm-thick Ti layer. The insertion of HfO_x_ and TiO_x_ yields an unstable interface. Using an active electrode material in combination, rather than the single structure of HfO_2_, improves its interfacial properties and stabilizes the switching characteristics of ReRAM due to the oxygen adsorption effect [35,36]. Figure 6b shows that the oxide layer is made of 6 nm-thick HfO_2_ and 4 nm-thick Al_2_O_3_ and embodies a multilayer structure. This structure improves the uniformity of the switching voltage and dispersion of HRS resistance [49]. Additionally, inserting a thin layer of Al_2_O_3_ between the HfO_2_ layer and electrode reduces data retention and operating current [50]. Figure 6c depicts a tri-layer structure with a thin layer of Al_2_O_3_ inserted between the HfO_2_ layers. This structure improves the uniformity and stability of switching and allows for the precise control of conductive filaments [38,51].

The compliance current for ReRAM devices is set at 1 μA. Figure 7 shows the I–V characteristics according to the compliance current. A lower compliance current implies that the reset process starts at a lower voltage because the filament is not sufficiently formed. Additionally, a higher compliance current indicates that the filament is not fully ruptured and does not reform, meaning that the set/reset process does not occur. The voltage is swept from 3V to −3V. The I–V characteristics for the voltage sweep are shown in Figure 8. Early TiN/Ti/HfO_2_/TiN devices are formed at 2.85 V. Over several cycles of voltage sweeps, the SET voltage occurs in the range of 2.1 to 2.5 V, and the RESET voltage appears in the range of −1.5 to −2.9 V. HfO_2_-Al_2_O_3_ based device shows improved uniformity at RESET, while TiN/Ti/HfO_2_/TiN exhibits scattered switching characteristics over multiple cycles.

The I–V characteristic graph indicates that after the first RESET, the second SET occurs at a lower voltage than the first SET. This result implies the presence of filaments left over from the formation and rupture of conductive filaments. Figure 9 depicts changes in the conductive filament during a voltage sweep.

Figure 9a shows the initial state with almost no conductive filaments. Figure 9b demonstrates the state after the first SET; here, the input voltage exceeds the threshold value, and a conductive filament is formed to connect the upper and lower electrodes. Figure 9c shows the state after the first RESET, illustrating that a large number of conductive filaments collapsed during the RESET process, while some remained. Figure 9d represents the second SET process, which results in a different shape due to the remnants of conductive filaments. Figure 9 illustrates the migration of oxygen vacancies and filaments. Figure 10 demonstrates the change in conductance for an input pulse with an amplitude of 2.5 V and a period of 2 ms in TCAD.

Figure 11a shows changes in conductance for the same input pulse. Figure 11a(i–iii) demonstrate the conductance change for the TiN/Ti/HfO₂/TiN structure, the conductance change for the TiN/Ti/HfO₂/TiN structure, and the TiN/Ti/HfO₂/Al₂O₃/HfO₂/TiN structure, respectively. In Figure 11a, the multi-level states of conductance are shown by averaging the conductance from cycle to cycle. In addition, the coordination of pulses can create multi-level states. When the same pulse is employed, the TiN/Ti/HfO_2_/TiN structure shows a gradual increase and a sharp change in conductance. The TiN/Ti/HfO_2_/Al_2_O_3_/TiN structure exhibits a sharp increase and a gradual decrease, while the TiN/Ti/HfO_2_/Al_2_O_3_/HfO_2_/TiN structure demonstrates a gradual increase and decrease. The SET–RESET voltage obtained from TCAD simulation, change in conductance over time, and normalized A values yielded from the MATLAB fitting are applied to NeuroSim simulation. The conductance of each structure is averaged over 30 random cycles. The deviations of cycle-to-cycle are reflected in the NeuroSim simulation.

A neural network trained using randomly selected images from a training dataset (60,000 images) and a test dataset (10,000 images) is classified. Figure 11b shows the results of a simulation with Epoch set to 100. Each device is verified for accuracy using NeuroSim. The TiN/Ti/HfO_2_/TiN structure yields an average accuracy of 92.73% and the highest accuracy of 94.35%. The TiN/Ti/HfO_2_/Al_2_O_3_/TiN structure exhibits an average accuracy of 94.27% and the highest accuracy of 95.43%. The TiN/Ti/HfO_2_/Al_2_O_3_/HfO_2_/TiN structure demonstrates an average accuracy of 94.56 and the highest accuracy of 95.76%.

Figure 12 illustrates the I–V switching characteristics with a tunneling process. During a SET event, the current behavior is more nonlinear than the I–V characteristic in Figure 8, resulting in the change in conductance. The high temperature causes non-uniformity and diffusion in filament growth and also promotes the TAT process. Therefore, it should be included to obtain realistic I–V characteristics. Figure 13 shows an example of the temperature-dependent I–V characteristics in the HfO₂/Al₂O₃ structure. SET and RESET occur at lower voltages as temperature increases because the filament formation and rupture become more active at higher temperatures. Additionally, tunneling effects are more pronounced at elevated temperatures.

Figure 14a displays the change in conductivity in a simulation with tunneling. The conductance sharply increases in the potentiation and gradually decreases in a depression. Each conductance is the average of the data over 30 cycles. The accuracy verification of these results is shown in Figure 14b. The TiN/Ti/HfO_2_/TiN structure yields an average accuracy of 84.47% and the highest accuracy of 87.71%. The TiN/Ti/HfO_2_/Al_2_O_3_/TiN structure exhibits an average accuracy of 86.03% and the highest accuracy of 89.73%. The TiN/Ti/HfO_2_/Al_2_O_3_/HfO_2_/TiN structure demonstrates an average accuracy of 87.99% and the highest accuracy of 90.37%. This represents a decrease of approximately 4–7%, compared to the accuracy shown in Figure 11b. This is due to the fact that the conductance changes due to tunneling are more asymmetric than the ideal case.

Inconsistent conductance changes make it difficult to reach the target conductance with the same pulse, reducing the convergence rate during the learning process. Nonlinearity and asymmetry affect learning accuracy. Nonlinearity causes more training accuracy loss. However, the bidirectional symmetric incremental conductance change has been found to maintain good accuracy even with relatively high nonlinearity. As the number of cycles increases, endurance gradually decreases. Additionally, with more cycles, functional reliability metrics such as dynamic range, nonlinearity, and asymmetry gradually deteriorate, leading to reduced accuracy [52].

## 6. Conclusions

In this paper, we presented the characterization of HfO_2_-based ReRAMs using Sentaurus TCAD. The switching behavior of the ReRAM is implemented through the KMC model, accounting for the application of the TAT model to describe the real device. The TCAD simulation performs the insertion of Al_2_O_3_ in the device, which is shown to display improved switching behavior. The device characteristics extracted from TCAD are used to evaluate the impact of ReRAM configuration on neuromorphic computing in NeuroSim. An idealized ReRAM would have high accuracy in a neuromorphic computing system. However, simulations with added tunneling show relatively low accuracy, implying that the accuracy of the conductance change depends on the asymmetric shape. As a result, TCAD can flexibly simulate various structures and processes, and characterize the properties of real devices. In support of TCAD simulation, an efficient verification of various ReRAM devices for neuromorphic computing can be achieved, enabling the acceleration ofthe pace of research by quickly finding ways to optimize device design.

## Figures and Tables

**Figure 1 nanomaterials-14-01864-f001:**
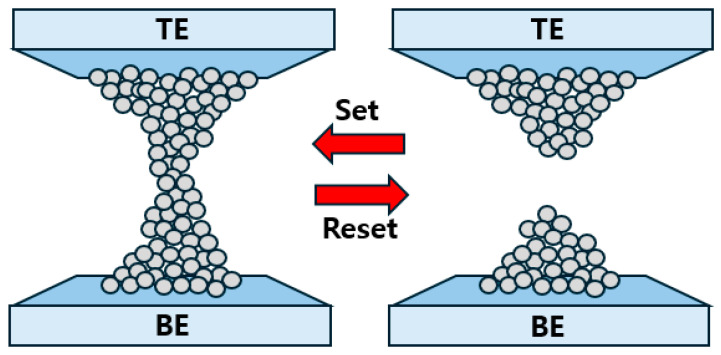
Operating principle of ReRAM. Set and Reset represent the formation and rupture of the filament, respectively.

**Figure 2 nanomaterials-14-01864-f002:**
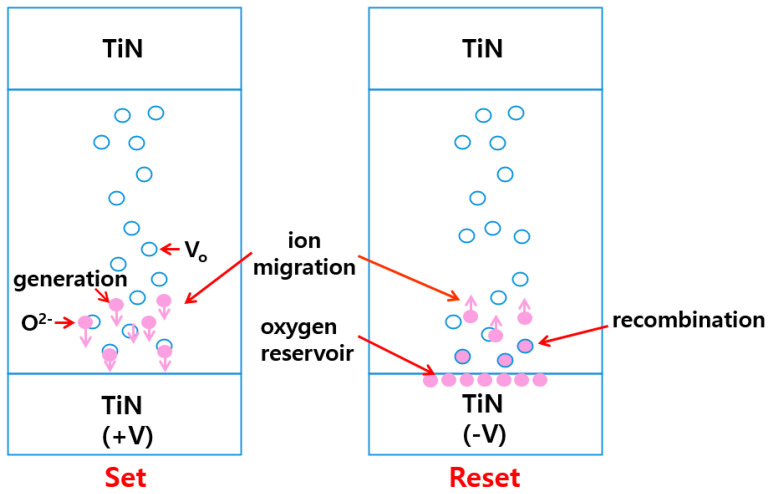
Schematic of the SET–RESET mechanism of metal oxide-based ReRAM.

**Figure 3 nanomaterials-14-01864-f003:**
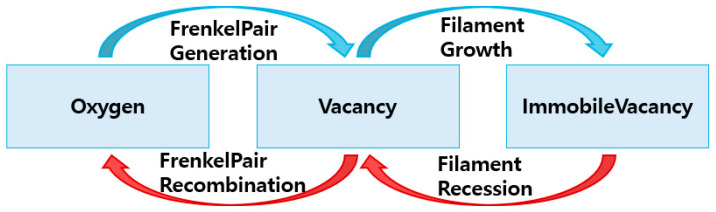
Formation and rupture mechanisms of conductive filaments using the KMC event.

**Figure 4 nanomaterials-14-01864-f004:**
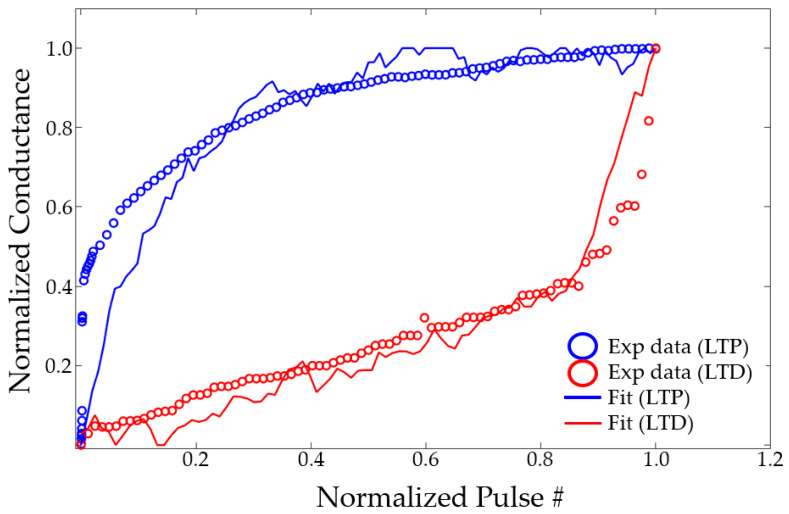
Fitting of ReRAM weight-update data. Normalized conductance and number of pulses is fitted with *A* values.

**Figure 5 nanomaterials-14-01864-f005:**
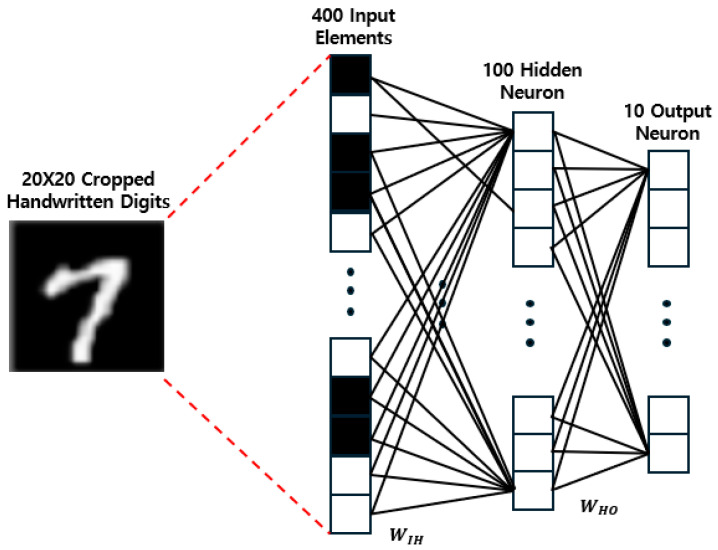
Two-layer MLP neural network for MNIST.

**Figure 6 nanomaterials-14-01864-f006:**
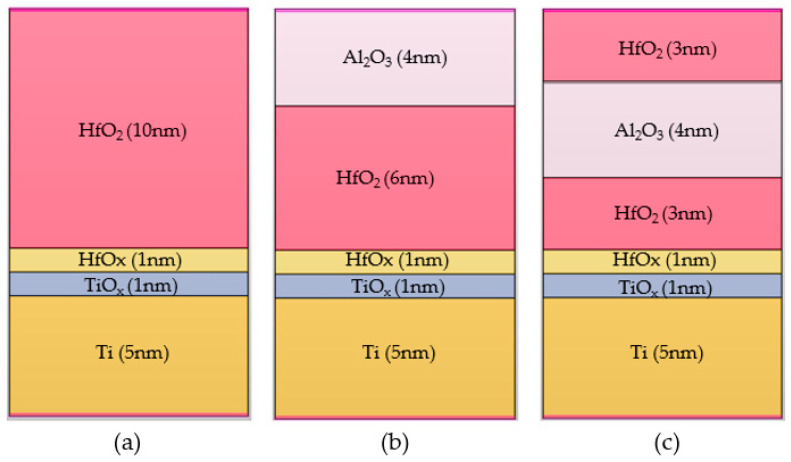
ReRAM structures fabricated with TCAD: (**a**) TiN/Ti/HfO_2_/TiN, (**b**) TiN/Ti/HfO_2_/Al_2_O_3_/TiN, and (**c**) TiN/Ti/HfO_2_/Al_2_O_3_/HfO_2_/TiN. The top and bottom electrode TiN are not shown in simulation.

**Figure 7 nanomaterials-14-01864-f007:**
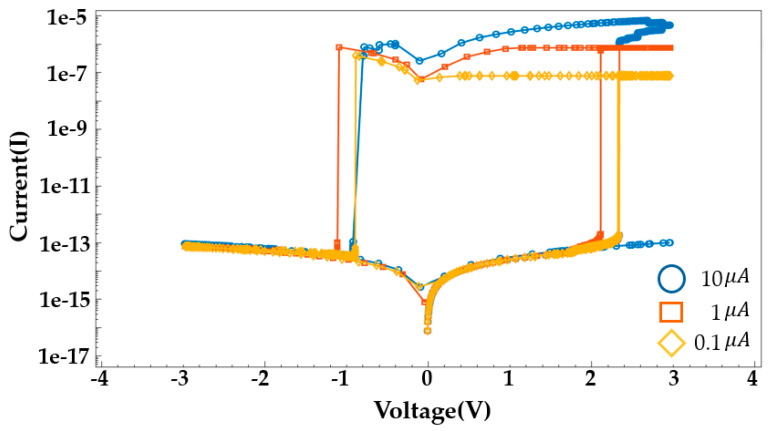
I–V characteristics according to the compliance current.

**Figure 8 nanomaterials-14-01864-f008:**
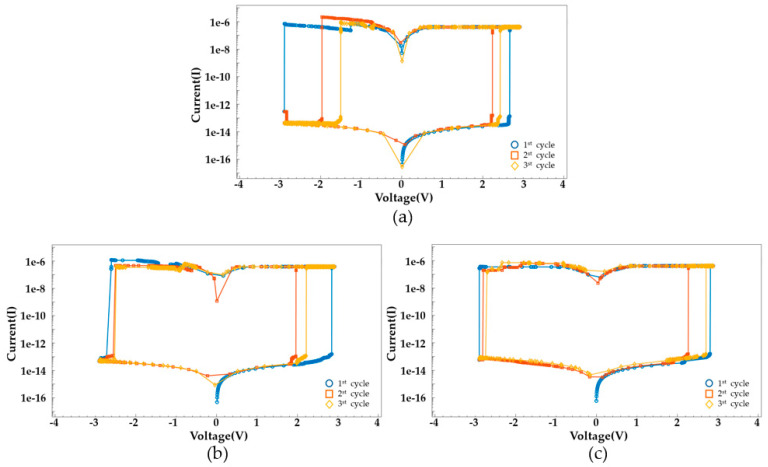
Ideal I–V switching characteristics for different cycles of (**a**) TiN/Ti/HfO_2_/TiN, (**b**) TiN/Ti/HfO_2_/Al_2_O_3_/TiN, and (**c**) TiN/Ti/HfO_2_/Al_2_O_3_HfO_2_/TiN.

**Figure 9 nanomaterials-14-01864-f009:**
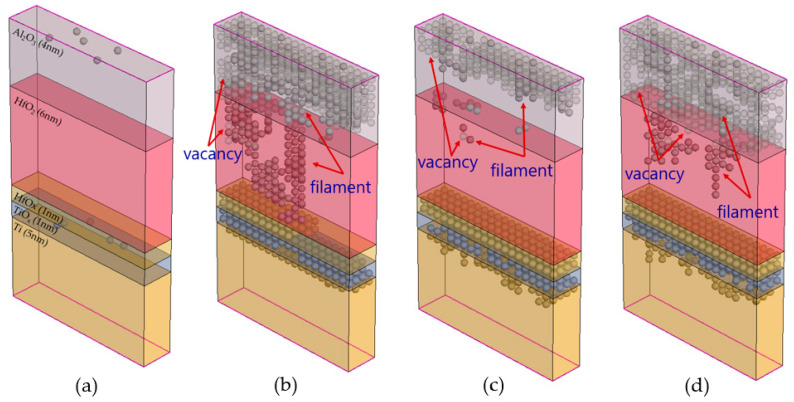
Changes in the conductive filament and oxygen vacancies: (**a**) initial state, (**b**) after the first SET, (**c**) after the first RESET, and (**d**) after the second SET.

**Figure 10 nanomaterials-14-01864-f010:**
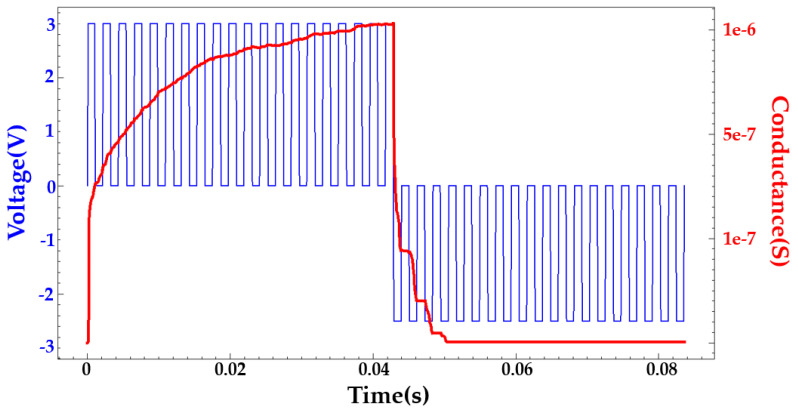
The change in conductance over the identical pulse in TCAD. The positive and negative voltages are 3 V and −2.5 V, respectively.

**Figure 11 nanomaterials-14-01864-f011:**
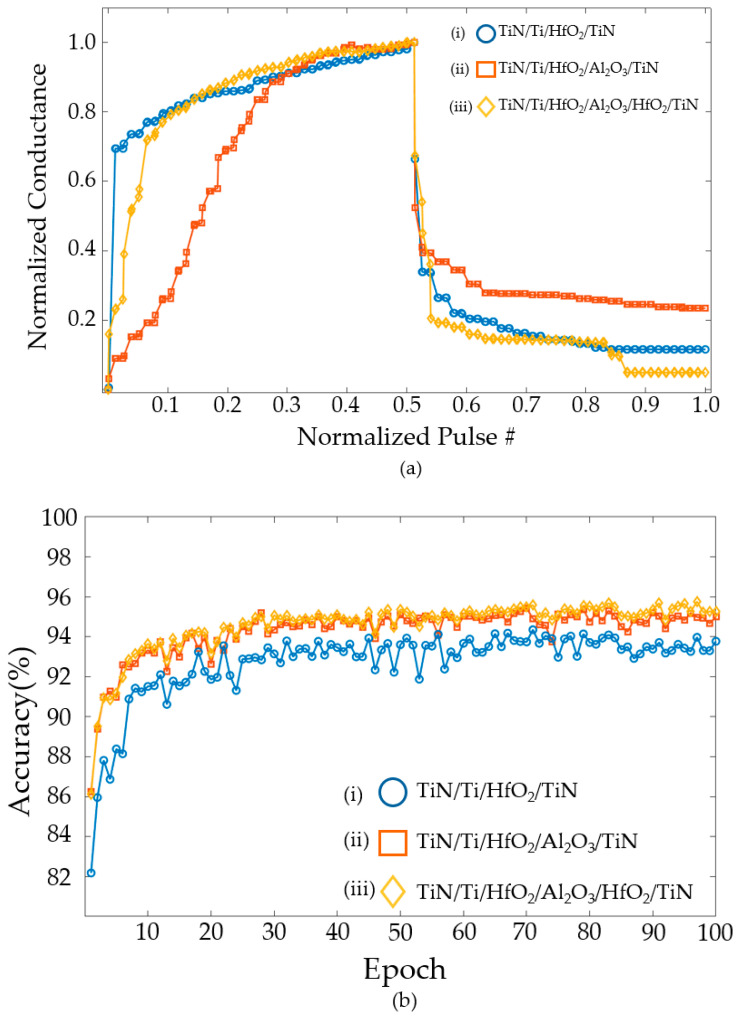
(**a**) Conductance changes and (**b**) accuracy for (**i**) TiN/Ti/HfO_2_/TiN, (**ii**) TiN/Ti/HfO_2_/Al_2_O_3_/TiN, and (**iii**) TiN/Ti/HfO_2_/Al_2_O_3_/HfO_2_/TiN structures.

**Figure 12 nanomaterials-14-01864-f012:**
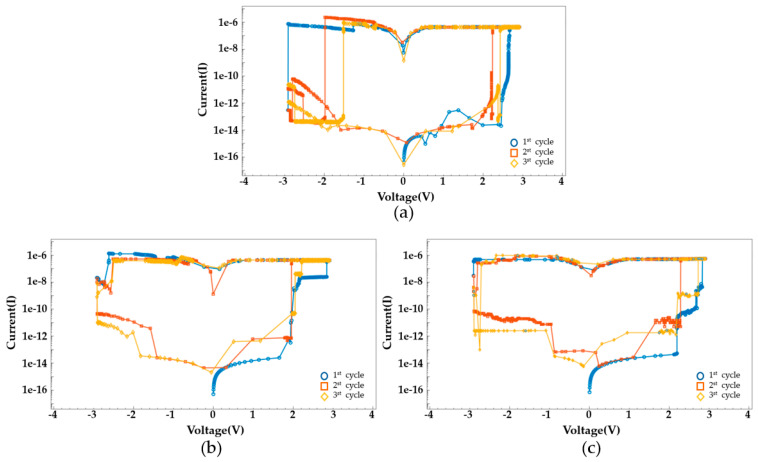
I–V switching characteristics with TAT for different cycles of (**a**) TiN/Ti/HfO_2_/TiN, (**b**) TiN/Ti/HfO_2_/Al_2_O_3_/TiN, and (**c**) TiN/Ti/HfO_2_/Al_2_O_3_HfO_2_/TiN.

**Figure 13 nanomaterials-14-01864-f013:**
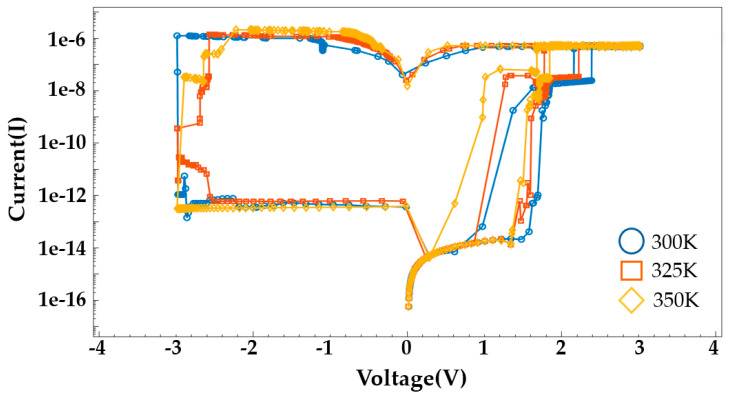
The temperature-dependent I–V characteristics in the HfO₂/Al₂O₃ structure.

**Figure 14 nanomaterials-14-01864-f014:**
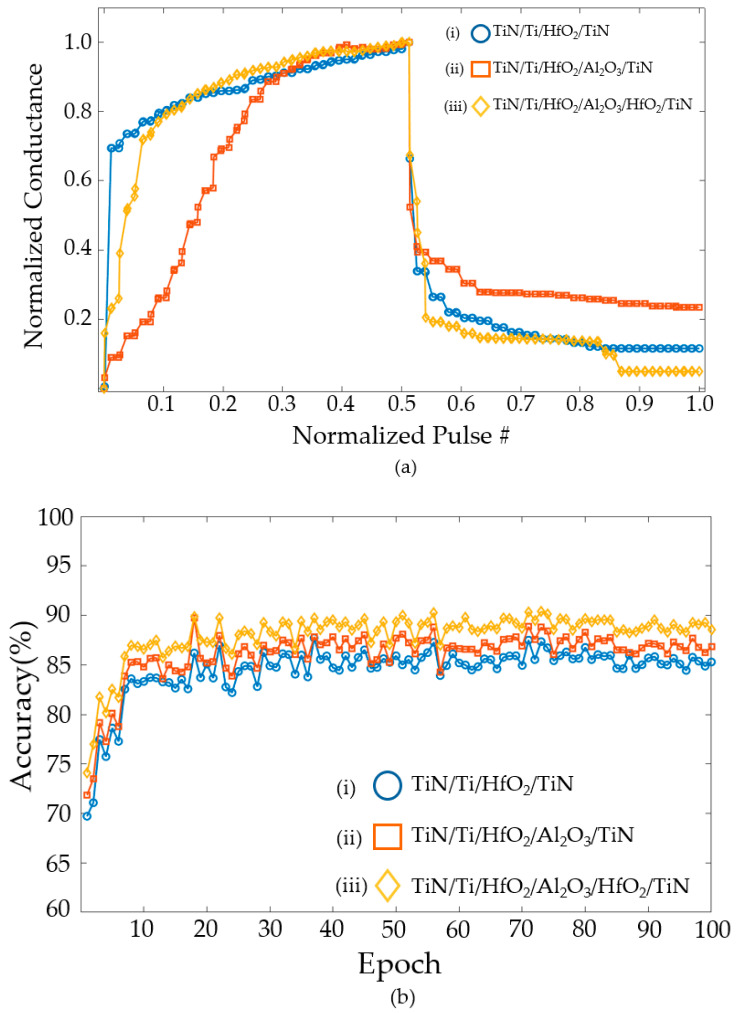
(**a**) Conductance change and (**b**) accuracy for (**i**) TiN/Ti/HfO_2_/TiN, (**ii**) TiN/Ti/HfO_2_/Al_2_O_3_/TiN, and (**iii**) TiN/Ti/HfO_2_/Al_2_O_3_/HfO_2_/TiN structures with TAT.

**Table 1 nanomaterials-14-01864-t001:** Physical event support.

Physical Event	Keywords	Relevant Equation
Diffusion	Diffusion	$A(r_{1})\to A(r_{2})$
Bulk generation or recombination	Generation, Recombination	$0\Leftrightarrow A\left(r\right)+B(r)$
Interface generation or recombination	Generation, Recombination	$0\Leftrightarrow A\left(r_{1}\right)+B(r_{2})$
Filament growth or recession	Filament growth, Filament recession	F(r)⇔A(r)

**Table 2 nanomaterials-14-01864-t002:** Device characterization parameters.

	HfO_2_	Al_2_O_3_
**Dielectric constant**	25 [43]	9 [43]
**O** **ptical permittivity**	3.74 [44]	3.43 [45]
**Band-gap (eV)**	5.4 [46]	8.8 [47]
**Activation energy (eV)**	4.8 [48]	1.8 [48]

## Data Availability

The data present in this study are available on request from the corresponding author.

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
