# Peer review of "TCAD Simulation of Resistive Switching Devices: Impact of ReRAM Configuration on Neuromorphic Computing"

_nanomaterials, 2024, doi:10.3390/nano14231864_

Round 1

Reviewer 1 Report

Comments and Suggestions for Authors

The manuscript (nanomaterials-3305047), TCAD Simulation of Resistive Switching Devices: Impact of ReRAM Configuration on Neuromorphic Computing, shows the results of the ReRAM architecture impact on device characterizations and potential neuromorphic computing applications. Authors presented quite comprehensive analysis in this work. However, some technical comments still need to be addressed before further confirm the outcomes, shown as following - 

1. Please provide a benchmark Table to review the proposal structures and actually device characterizations in the real world as compared to the TCAD simulation results. Does that the simulated results match truly device characterizations? 

2. For the DC IV behaviors in Figure 7, most of the devices are sharply change the resistance in both SET and RESET region. However, in the Figure 10, the device can have middle states by pulse. Can authors provide the device-to-device and cycle-to-cycle studying, retention studying, and multi-level states demonstrated on the DC I-V as well with retention to see how to connect from Figure 7 to Figure 10?

3. The tunneling process added on the Figure 12 is not fully explain in the manuscript. Did authors check the temperature dependence and know that there are no temperature-related current transport behaviors happened in Figure 12? Also, please provide the device-to-device and cycle-to-cycle studying, retention studying, and multi-level states demonstrated on the DC I-V as well with retention to connect from Figure 12 to Figure 13?

4. Can authors provide the reliability degraded case studying in this work to see how the endurance impact on the neuromorphic properties? 

Due to the above comments, this referee would like to put the manuscript status as "Major Revision" in the current phase. 

Reviewer 2 Report

Comments and Suggestions for Authors

This paper shows TCAD simulation results of ReRAM devices for neural network applications.

The authors first explained how they implemented the ReRAM in the TCAD software Synopsys. Then, the neural network was simulated using NeuroSim, a previously published model. The parameters of the NeuroSim model were extracted from the TCAD results, so the originality lies in the TCAD implementation, and the extraction of the NeuroSim parameters.

I found the paper very interesting to read, with most of the basic concepts very well explained.

However, the implementation part is very specific to the Synopsys software, and may be of limited interest for someone not familiar with it.

I have the gollowing propositions to help improve the paper:

1) In Eq. (2), an optical permittivity was defined. Is it the same as the relative permittivity ? What is the value of this parameter in the simulations ?

2) The heat equation is considered in (3). Does this means that it is necessary to take the temperature inhomogeneity and diffusion into account ? If this is the case, why ?

3 ) It should be specified if Fig. 9 is the result of TCAD simulations, or a plot of Eqs. (4-6).

4) About the tunneling effect considered in Figs. 12-14, is it TAT tunneling ? Please specify it in the text.

5) Since the formation of the conductive filament is a random process, the transition voltages  between the low and high resistive states, shown in Fig. 7 and 12, should not always be exactly the same. This should cause some additional variability between each of the ReRAM device of the neural network, and also in the extracted NeuroSim parameters from the TCAD results . Could the authors comment on that?

Reviewer 3 Report

Comments and Suggestions for Authors

This manuscript reports on modeling ReRAMs using TCAD and checks its accuracy for neuromorphic systems. Kinetic Monte Carlo is employed to model the ReRAM’s switching behaviors. To make the ReRAM characteristics realistic, the authors used a trap-assisted tunneling model and thermal equations. ReRAM made with HfO2-Al2O3 showed better-switching behaviors than ReRAM made with HfO2. The conductance varied based on the ReRAM’s structure, whereas their TCAD data were tested in the neuromorphic system using the MNIST dataset. My recommendation is a major revision to address the following issues.

1. When introducing the significance of developing memory devices that utilize new operating principles, innovative structures, and advanced materials (p.1, lines 25-26), it’s beneficial to highlight recent groundbreaking achievements in the field. For instance, Merces et al. recently reported on the creation of ultraflexible origami tessellations with shape-memory properties that can be integrated with electronics (doi.org/10.1002/adma.202313327), and Wang et al. introduced an organic electrochemical device capable of multi-modal sensing, memory, and processing (doi.org/10.1038/s41928-023-00950-y).

2. Is "memristor" a compound word for a memory device and (1) a transistor or (2) a resistor? Please clarify this aspect in the revised manuscript.

3. The authors asserted that NCS research has primarily concentrated on memristor devices and crossbar structures (p. 1, lines 33-34). However, this perspective only partially captures the current advancements in NCS development. It is recommended to introduce the crucial role of 3-terminal devices, such as transistors, in the evolution of today’s advanced neuromorphic systems too. For the authors' reference, a recent review (doi.org/10.1021/acs.chemrev.4c00369) provides compelling insights into both 2- and 3-terminal synaptic devices, while another review (doi.org/10.1002/advs.202305611) effectively summarizes the most promising breakthroughs in transistor-based artificial synapses. Such inclusions are necessary for a fair and well-rounded introduction to the field.

4. What is the energy consumption per bit in the ReRAM system? Is it comparable to the energy consumption achieved by Liu et al. (https://doi.org/10.1002/adfm.202200959)?

5. By defining the vacancies as voids by "Particles2," does it allow the vacancies to migrate in the TCAD model?

6. Although the rate- and heat equations are well-known, I recommend citing the source articles as references in the revised manuscript.

7. In Figure 4, why do the fits provide coarse adjustments at the normalized conductance's high-variation regions?

8. How does using current compliance affect the device switching properties? Do different compliance levels result in varying properties?

9. For conciseness, Figures 9-11 could be rearranged in a 3-panel revised figure, whereas Figures 13,14 could be rearranged in a 2-panel revised figure.

10. Minor points: There are a few grammars to be corrected (e.g., "TCAD 'are' widely used" in p.3 line 108; "Immobile Vaca'm'cy" in Figure 3; the capitalized "The 'O'xygen" in p.4 line 128;

Reviewer 4 Report

Comments and Suggestions for Authors

The scientific content of this manuscript is limited, with nearly half of the text dedicated to popular science, which can be found in detail in the user manuals of TCAD and NeuroSim, as well as in review articles on ReRAM. In the experimental section, merely tweaking the structural parameters from TCAD examples, extracting the data, and directly running it in NeuroSim does not require much time and lacks significant innovation. Therefore, I do not recommend publication.

Author Response

In this paper, a new method of the ReRAM device model based on TCAD simulation is newly presented. This methodology can flexibly simulate various structures and processes and characterize the properties of real ReRAM devices. In support of TCAD simulation, an efficient verification of various ReRAM devices for neuromorphic computing can be achieved, enabling to accelerate the pace of research by quickly finding ways to optimize ReRAM device design.

Round 2

Reviewer 1 Report

Comments and Suggestions for Authors

Authors have replied to this referee in detail. No furthe rcomments from this referee. 

Reviewer 3 Report

Comments and Suggestions for Authors

The authors have clearly responded to all my concerns, and the manuscript has been improved as well. I recommend accepting it for publication in Nanomaterials.

Reviewer 4 Report

Comments and Suggestions for Authors

All other issues in the manuscript have been addressed, and I have no further suggestions to make.